# Rapid detection of myeloid neoplasm fusions using single-molecule long-read sequencing

Olga Sala-Torra[1,2], Shishir Reddy[2], Ling-Hong Hung[2], Lan Beppu[1], David Wu[3], Jerald Radich[1,3], Ka Yee Yeung[2], Cecilia C. S. Yeung[1,3]*

1 Translational Science and Therapeutics Division, Fred Hutchinson Cancer Center, Seattle, Washington, United States of America, 2 University of Washington, Seattle, Washington, United States of America, 3 School of Engineering and Technology, University of Washington Tacoma, Tacoma, Washington, United States of America

* cyeung@fredhutch.org

## Abstract

Recurrent gene fusions are common drivers of disease pathophysiology in leukemias. Identifying these structural variants helps stratify disease by risk and assists with therapy choice. Precise molecular diagnosis in low-and-middle-income countries (LMIC) is challenging given the complexity of assays, trained technical support, and the availability of reliable electricity. Current fusion detection methods require a long turnaround time (7–10 days) or advance knowledge of the genes involved in the fusions. Recent technology developments have made sequencing possible without a sophisticated molecular laboratory, potentially making molecular diagnosis accessible to remote areas and low-income settings. We describe a long-read sequencing DNA assay designed with CRISPR guides to select and enrich for recurrent leukemia fusion genes, that does not need a priori knowledge of the abnormality present. By applying rapid sequencing technology based on nanopores, we sequenced long pieces of genomic DNA and successfully detected fusion genes in cell lines and primary specimens (e.g., *BCR*::*ABL1*, *PML*::*RARA*, *CBFB*::*MYH11*, *KMT2A*::*AFF1*) using cloud-based bioinformatics workflows with novel custom fusion finder software. We detected fusion genes in 100% of cell lines with the expected breakpoints and confirmed the presence or absence of a recurrent fusion gene in 12 of 14 patient cases. With our optimized assay and cloud-based bioinformatics workflow, these assays and analyses could be performed in under 8 hours. The platform's portability, potential for adaptation to lower-cost devices, and integrated cloud analysis make this assay a candidate to be placed in settings like LMIC to bridge the need of bedside rapid molecular diagnostics.

## Introduction

Low-and middle-income countries (LMICs) suffer the heaviest burden of cancer deaths. Scarce pathology services that result in incomplete, inappropriate, or delayed diagnoses are one of the causes of higher morbidity and mortality rates. A slow or incorrect diagnosis can result in disease progression with worsening prognosis, or incorrect treatment decisions [1–4]. A systematic review showed that earlier testing after the start of symptoms is associated with

**Data Availability Statement:** All relevant cell line and de-identified patient data are available from the NCBI SRA repository with accession number PRJNA999047 (https://www.ncbi.nlm.nih.gov/

bioproject/PRJNA999047). All identifiers for patient specific data files have been removed prior to SRA submission.

**Funding:** CY is supported by NCCN young investigator award and the Hyundai Hope on Wheels Scholars Award. CY and OST are also supported by R21 CA280520. JR is supported by R01 CA175008-06, UG1 CA233338-02, and R01 CA175008-10, National Institutes of Health, Bethesda, MD. LHH, SR, and KYY are supported by NIH grants R21CA280520, R01GM126019 and U24HG012674. SR is also supported by the Vicky L. Carwein and William B. Andrews Endowments for Graduate Programs. The funders had no role in study design, data collection and analysis, decision to publish, or preparation of the manuscript.

**Competing interests:** I have read the journal's policy and the authors of this manuscript have the following competing interests: LHH and KYY have equity interest in Biodepot LLC, which receives compensation from NCI SBIR contract numbers 75N91020C00009 and 75N91021C00022. The terms of this arrangement have been reviewed and approved by the University of Washington in accordance with its policies governing outside work and financial conflicts of interest in research. This work is related to the following two provisional patent applications: "Optimized assay and efficient computational workflow to deliver same-day leukemia diagnostics" (patent application 63/416,888) "Detecting Leukemia Specific Gene Fusions" (patent application 63/309261)

lower-stage disease and improved survival benefits for breast, colorectal, head and neck, prostate cancers and melanoma [5]. In 2017, only 1/4 of low-income countries reported having readily available access to pathology services [6]. In response, the WHO published a guide to early cancer diagnosis emphasizing the importance of early diagnosis to include access to disease evaluation to guide subsequent treatment [5].

The current classification of myeloid malignancies is largely based on molecular and genetic aberrations [7, 8]. Recurrent gene rearrangements are present in 30–40% of acute myeloid leukemia (AML), and well-described driver fusions sometimes suffice to diagnose leukemias (for example, *PML::RARA, or RUNX1::RUNX1T1 among others)* [7]. These fusion genes confer specific clinical and biological characteristics as drivers of leukemogenesis that can assist in prognosis stratification and inform treatment decisions. Detection of diagnostic fusions may be especially relevant in developing countries to guide the use of resources as they have several novel target therapies available since they are included in the WHO list of essential medicines. Examples include tyrosine kinase inhibitors (TKI) in chronic myelogenous leukemia (CML) [9] that block BCR::ABL1 kinase activity and differentiation therapy with all-trans retinoic acid (ATRA) in acute promyelocytic leukemia (APL) targeting the (*PML::RARA*) fusion [10, 11].

Typically, fusion genes are detected by fluorescence *in situ* hybridization (FISH), polymerase chain reaction (PCR) assays, or next-generation sequencing (NGS). NGS is an attractive technology because it should be able to detect a variety of genetic mutations found in leukemia, including fusions, insertions, deletions, and point mutations in oncogenic genes. However, conventional NGS has technical hurdles (short sequencing reads, introduction of false positive errors due to library preparation) and hardware issues (expensive machines, complicated bioinformatics). These issues make NGS hard to employ readily in LMIC settings.

The advancement of long-read sequencing technologies has enabled the sequencing of continuous single DNA or RNA molecules up to tens to hundreds of kilobases (kb) long [12]. Ongoing improvements in Nanopore sequencing accuracy have reduced error rates to less than 5% [13] but remain higher than those for Illumina and Ion Torrent platforms, which are used frequently in clinical laboratories [14]. Long-read sequencing has made an impact on the understanding of the pathobiology of various diseases [15, 16], and its impact should increase as the sequencing quality improves and becomes more accurate [13, 17]. Addition of CRISPR--Cas9 for targeted enrichment concentrates the regions of interest prior to sequencing by Nanopore [18] without requiring amplification steps, thus optimizing sequencing time and efficiency. More importantly, the portability and affordability of the Nanopore sequencer MinION and Flongle hold great promise to impact the clinical field, especially in the LMIC setting. However, a major limitation has been the lack of analytical software featuring standardized parameters to aid in translation into clinical diagnostics. Implementing such a portable platform with reduced cost and cloud-based on-demand analysis workflow in developing countries would enable testing centers to provide sequencing without investing in a computing or bioinformatics infrastructure, thus bridging the gap between diagnosis and tailored treatment administration.

Here we report our success in developing an amplification-free CRISPR-Cas9 targeted enrichment sequencing protocol using Nanopore MinION flow cell and Flongles to detect fusions relevant to the diagnosis and classification of CML and AMLs. The ONT (Oxford Nanopore Technologies) Flongle is an adaptor for the MinION that provides cost-effective (~USD $90 per MinION flow cell) real-time sequencing for smaller assays of limited target genes. Our assays were designed to capture various breakpoints of CML and APL, as well as fusion genes resulting from inv(16) (*CBFB::MYH11*) and t(4;11)(*KMT2A::AFF1*). Simultaneous interrogation of these targets is a first step to rapid characterization of AMLs in a single assay combining data that up to now required multiple different techniques and providing

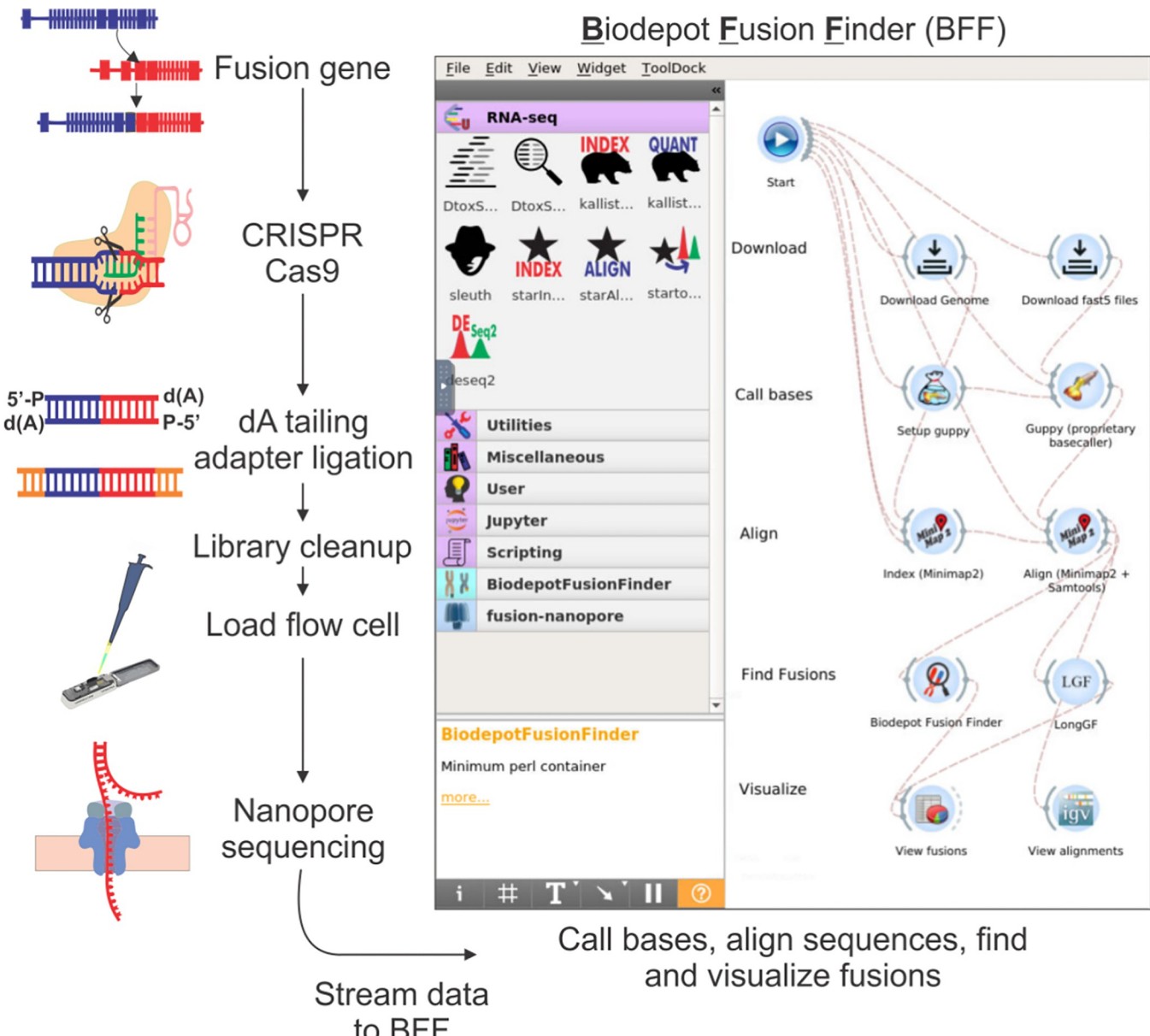

**Fig 1. Chemistry and bioinformatics workflow of rapid single-molecule long-read sequencing.** Genomic DNA may contain a target fusion gene. The CRISPR-Cas9 system binds via specific guideRNA (gRNA) designed to enrich for DNA containing regions of interest. The library preparation does not undergo any amplification and requires dA tailing, adapter ligation, and a clean-up before being loaded onto a sequencing flow cell. On the flow cell, libraries are sequenced by nanopores. The Biodepot-workflow-builder (Bwb) platform takes the input sequencing data in FAST5 format, performs quality control, base calling, and alignment. After alignment, different fusion finder tools were applied, including LongGF and the Biodepot Fusion Finder (BFF), before visualizing the results using the Integrative Genomics Viewer (IGV) for confirmation and interpretation.

relevant information promptly. In addition to this amplification-free CRISPR-Cas9 nanopore assay, we extended our previously developed cloud-based Nanopore data analysis pipeline [19] to include fusion detection and develop a custom breakpoint detection tool (see **Fig 1**). Using our optimized assay and our custom breakpoint finder, we showed that we can reliably detect and confirm fusion breakpoints in 80% of our specimens in under 3 hours total time including sequencing and data analysis.

## Materials and methods

### Cells lines and patient samples

Our assay was optimized using six cell lines: three with the *BCR*::*ABL1* fusion (K562, KU812, and KCL22), and NB4, MV4;11 and ME-1 that bear the *PML*::*RARA*, *KMT2A*::*AFF1*, *and CBFB*::*MYH11* fusions respectively. Residual mononuclear cells from primary specimens (six specimens from 5 patients with CML, six specimens from 5 patients with suspected APL, and two acute myeloid leukemia, not acute promyelocytic leukemia) were isolated using Ficoll reagent (Millipore-Sigma) and banked in liquid nitrogen until the time of the experiment. All specimens had been originally tested in a CLIA-certified laboratory according to standard clinical protocols [20]. IRB coverage was obtained for the use of residual laboratory samples. Patient samples were de-identified to the Nanopore testing lab, and cytogenetic or molecular results were confirmed after nanopore results were rendered. Characteristics and demographics of specimens and patients are listed in **Table 1**.

### Library preparation and sequencing assay

For the cell lines and 11/14 patient specimens, the DNA was extracted with PureGene (Qiagen, Germantown, MD, USA) following the standard protocol. Special caution, including the use of wide-bore pipette tips and moderate centrifuge spin velocity, was exercised to minimize fragmenting DNA strands. Two DNA specimens were extracted with AllPrep DNA/RNA Kit (Qiagen, Germantown, MD, USA) and one with QiAgen X-tractor with Reagent Pack DX (Qiagen, Germantown, MD, USA). cRNA guides were designed to direct Cas9 to cut on the genomic proximity of each region involved in the translocations studied. When the target region was large, guides were tiled across the region to maximize coverage. Guides were

**Table 1. Characteristics of the primary specimens analyzed.**

| Patient code | Age | Gender | Diagnosis | Specimen Source | Disease Burden | Cytogenetics/Molecular |
|---|---|---|---|---|---|---|
| CML1 | 62 | Male | CML-AP Rlps | BM | 2% blasts | 46,XY,t(5;12)(q33;q15),t(9;22)(q34;q11.2)[2] / 47,sl,+8[18] |
| CML2 | 45 | Female | CML-BC | PB | NA | Outside cytogenetics confirmed t(9;22) but only in 3/21 karyotypes. Full karyotype not available. |
| CML3 | 56 | Male | CML-AP | PB | NA | Patient with long standing p190 CML. At this timepoint sample presents MDS/MPN with low level BCR::ABL1 by PCR. |
| CML4 | 21 | Male | CML-CP | PB | 3% blasts | 46,XX,t(9;22)(q34;q11.2)[20] |
| CML5 | 62 | Male | CML-AP Rlps | PB | 3% blasts | 46,XY,t(5;12)(q33;q15),t(9;22)(q34;q11.2)[2] / 47,sl,+8[18] |
| CML6 | 70 | Female | CML-BC | PB | 16% blasts | 46,XX,t(9;22)(q34;q11.2)[4]; Confirmed by molecular studies. |
| AML1 | 39 | Male | AML | PB | 90% blasts | 46,XY,t(9;11)(q21;q23)[20] |
| AML2 | 40 | Female | AML | PB | 20% blasts | 47,XX+8,inv(16)(p13.1q22)[17]/46,XX[3] |
| APL1 | 83 | Male | APL | PB | <1% blasts | Patient with collision multiple myeloma and APL in BM with 30% blasts, not available for analysis. No circulating APL in PB. |
| APL2 | 34 | Female | APL | BM | 80% blasts | 46,XX,der(2)t(2;17)(q33;q21)t(15;17)(q22;q21), der(15) t(15;17),der(17)t(2;17) [3] / 47,sl+8[3] / 48,sdl,+8[12] / 46,XX[2] |
| APL3 | 23 | Male | APL | PB | 75% blasts | NA |
| APL4 | 39 | Male | APL | BM | 83% blasts | AML with isochromosome 17q |
| APL5 | 39 | Male | APL | PB | 79% blasts | AML with isochromosome 17q |
| APL6 | 43 | Male | APL | BM | 22% blasts in a <5% marrow | 46, XY,t(1;7)(q21;q21), t(4;10)(q21;q25), add(8)(p23), del(9)(p22),t(11;17)(q23;q25) [4] / 46, XY[16] from 3 months prior |

CML = chronic myeloid leukemia; AML = acute myeloid leukemia; APL = acute promyelocytic leukemia; AP = accelerated phase, BC = blast crisis, PB = peripheral blood, BM = bone marrow; Rlps = relapsed; MDS/MPN = myelodysplastic/myeloproliferative neoplasm; NA = Not Available.

designed to capture *PML::RARA*, *BCR::ABL1* p210, *KMT2A::AFF1*, and *CBFB::MYH11*, including different fusion isoforms. Guides were designed using Chopchop (https://chopchop.cbu.uib.no/) with the CRISPR-Cas9 and nanopore enrichment settings previously described [21].

We used five micrograms of DNA as input for each cell line and 2 to 5 micrograms for primary specimens. The average DNA integrity number (DIN) was 9.2 (range: 7.5–9.8). **Fig 1** shows a schematic of our workflow, and details of the library prep are published [21]. Briefly, enrichment of target regions was obtained using Oxford Nanopore Technologies' "Targeted, amplification-free DNA sequencing using CRISPR-Cas9" protocol [18]. The different guides used in the assay were pooled in equimolar amounts of each guide. Through an initial dephosphorylation step, the 5' ends of the DNA becomes inaccessible to adapter ligation. Double-stranded DNA breaks that excise the region of interest are generated with the directional, target-specific RNA guides complexed with tracrRNA and Cas9 enzyme. The Cas9 complex remains bound to the 5' end of the guide, and the resulting new DNA ends contain a phosphorylated 5' end that is available for dA tailing and adapter ligation [18]. All libraries generated in this manner were run on a MinION (Oxford Nanopore Technologies, Oxford, UK) nanopore sequencer using MinION flow cells version 9.4 or Flongles. Modifications for libraries sequenced on the Flongle were only at the library loading step, in which the amount of Sequencing Buffer and library beads (both SQK-LSK 109, ONT) are reduced from 35 to 13 and 25.5 to 7.5UL respectively, and 0.5UL of SQT is added. QC parameters tracked for each run are listed in **Table 2**.

**Orthogonal confirmation of fusions and breakpoints.** Patient specific *BCR::ABL1* breakpoints were confirmed by performing PCR and sanger sequencing in 2 cases. Primers were designed using Primer3 v. 0.4.0 [22]. 100 ng of DNA were amplified, the PCR product was run on a 2% agarose gel, and Sanger sequenced to confirm the genomic breakpoint. All other patients' were confirmed with clinical karyotype and fluorescent in-situ hybridization data. BCR-ABL cell lines breakpoints were confirmed by published data [23].

**Table 2. Characteristics of the runs.**

| Sample | DIN | Device | GB of Data | Mean Coverage | All Reads | N50 All Reads | Median Read Length | Median PHRED | % Reads Aligned |
|--------|-----|--------|-----------|---------------|-----------|---------------|-------------------|--------------|-----------------|
| K562 | 9.8 | Flongle | 0.12 | 0.04 | 35943 | 5980 | 2200 | 9.52 | 87.57 |
| KU812 | 8.7 | Flowcell | 0.36 | 0.11 | 86579 | 9690 | 1820 | 12.76 | 96.29 |
| KCL22 | 9.7 | Flowcell | 0.69 | 0.21 | 50439 | 32200 | 6290 | 12.26 | 91.86 |
| NB4 | 9.1 | Flongle | 0.07 | 0.02 | 23122 | 5890 | 1450 | 9.13 | 88.86 |
| MV4;11 | 8.6 | Flowcell | 0.28 | 0.08 | 40000 | 27100 | 1890 | 11.98 | 89.34 |
| ME1 | 8.2 | Flowcell | 0.91 | 0.28 | 433846 | 16500 | 872 | 11.17 | 84.69 |
| CML1 | 9.8 | Flowcell | 0.92 | 0.28 | 32723 | 24600 | 9600 | 12.31 | 91.74 |
| CML2 | 9.6 | Flowcell | 0.04 | 0.01 | 4026 | 22400 | 6520 | 11.15 | 72.08 |
| CML3 | 9.8 | Flongle | 0.09 | 0.03 | 9857 | 19900 | 5860 | 9.45 | 85.54 |
| CML4 | 9.7 | Flongle | 0.93 | 0.28 | 109648 | 15200 | 1800 | 9.31 | 85.97 |
| CML5 | ND | Flongle | 0.04 | 0.03 | 3907 | 22400 | 8090 | 8.95 | 91.06 |
| CML6 | 9.2 | Flowcell | 5.00 | 1.52 | 381552 | 30800 | 7090 | 11.94 | 90.73 |
| AML1 | 9.3 | Flowcell | 5.47 | 1.66 | 639884 | 15100 | 6340 | 9.89 | 86.76 |
| AML2 | 9.2 | Flowcell | 1.59 | 0.48 | 161010 | 19400 | 6540 | 12.00 | 89.95 |
| APL1 | 9.6 | Flowcell | 0.17 | 0.05 | 13486 | 24700 | 7450 | 12.94 | 92.49 |
| APL2 | 7.7 | Flowcell | 1.1 | 0.33 | 40919 | 24200 | 8770 | 12.12 | 92.69 |
| APL3 | 9.7 | Flowcell | 0.31 | 0.09 | 13625 | 22500 | 6910 | 12.42 | 93.51 |
| APL4 | 9.6 | Flowcell | 1.63 | 0.49 | 176665 | 25200 | 3740 | 12.38 | 93.27 |
| APL5 | 9.3 | Flowcell | 0.22 | 0.07 | 31863 | 24200 | 2450 | 11.82 | 91.49 |
| APL6 | 7.5 | Flowcell | 1.06 | 0.32 | 485374 | 4650 | 1230 | 11.70 | 84.27 |

## Biodepot-workflow-builder: Interactive and accessible front end for fusion detection

We present a graphical, reproducible, and cloud-enabled fusion detection workflow consisting of all the steps of the analyses, including base calling, alignment, fusion detection, and visualization (**Fig 1**). In contrast to the NanoFG workflow from Stangle et al. [24], our platform includes the computationally intensive base calling step, an interactive graphical user interface, can readily leverage GPUs, and can be deployed on the cloud. Thus, analyses are fast, and our platform is easily and broadly accessible to users. We extended the Biodepot-workflow-builder (Bwb) [25] platform in which each computational task (module) is represented by a graphical widget that calls a software container in the back end. Software containers like Docker include all software dependencies and libraries required to execute the code. We have recently developed a Bwb workflow [19] to support the processing of Nanopore data that includes the use of base callers Guppy [26] version 6.4.6 using the r9.4.1 hac model, alignment using minimap2 and visualization of results using the Integrative Genomics Viewer (IGV) and GRCh37 hg19 as the reference genome. QC was performed using PycoQC [27]. Minimap2 [28] was used as the aligner and variant caller. Fusions are visualized on IGV [29] and confirmed on Blast. Most importantly, we extend our previous work by adding support for fusion detection, including LongGF [30] and our own custom software, "Biodepot Fusion Finder" (BFF).

## Bioinformatics pipeline for fusion detection: LongGF

LongGF [30] is a software tool for fusion detection optimized for the high base calling error rates and alignment errors commonly found in long-read sequencing data. LongGF takes as input a BAM file containing alignments (generated by minimap2 in this pipeline) and a GTF file containing the definitions of known genes. The output is a log file with detected gene fusions and their supporting reads. We created a graphical widget for LongGF in the Bwb. **Fig 2A** shows a screenshot of the comparative workflows with output for LongGF versus BFF.

## Custom fusion detection tool: Biodepot Fusion Finder (BFF)

Reads that span a fusion gene will align to coordinates in both parts of the fusion and provide a specific breakpoint coordinate. We wrote a new software tool, the Biodepot Fusion Finder (BFF) that examines alternate alignments for each read and identifies the ones that map to coordinates spanning a set of known breakpoints. This strategy is similar to the one used by LongGF, except that we allow for errors in the alignment near the breakpoint. The Breakpoint Finder identifies fusions not detected by LongGF, which looks for supporting reads where the breakpoints are exactly matched. The user can provide a panel of breakpoint coordinates of interest. If no panel is provided, BFF will return candidate fusions that span non-adjacent genomic regions. Incomplete coordinates for breakpoints in the panel are supported–the user does not need to define the exact coordinates, nor do both breakpoints need to be given. BFF will return all the reads that match the breakpoints panel in a text file for further inspection by the user if desired. BFF will identify the precise breakpoint in each read and also give a count of the number of breakpoints found nearby in all reads. The user can also provide a file with guide coordinates to obtain additional enrichment metrics. A containerized widget was developed that can be integrated with the Bwb workflows for processing nanopore data. Using these workflows, we can directly analyze raw nanopore data and obtain lists of candidate fusions.

## A: Bwb workflow with LongGF &BFF

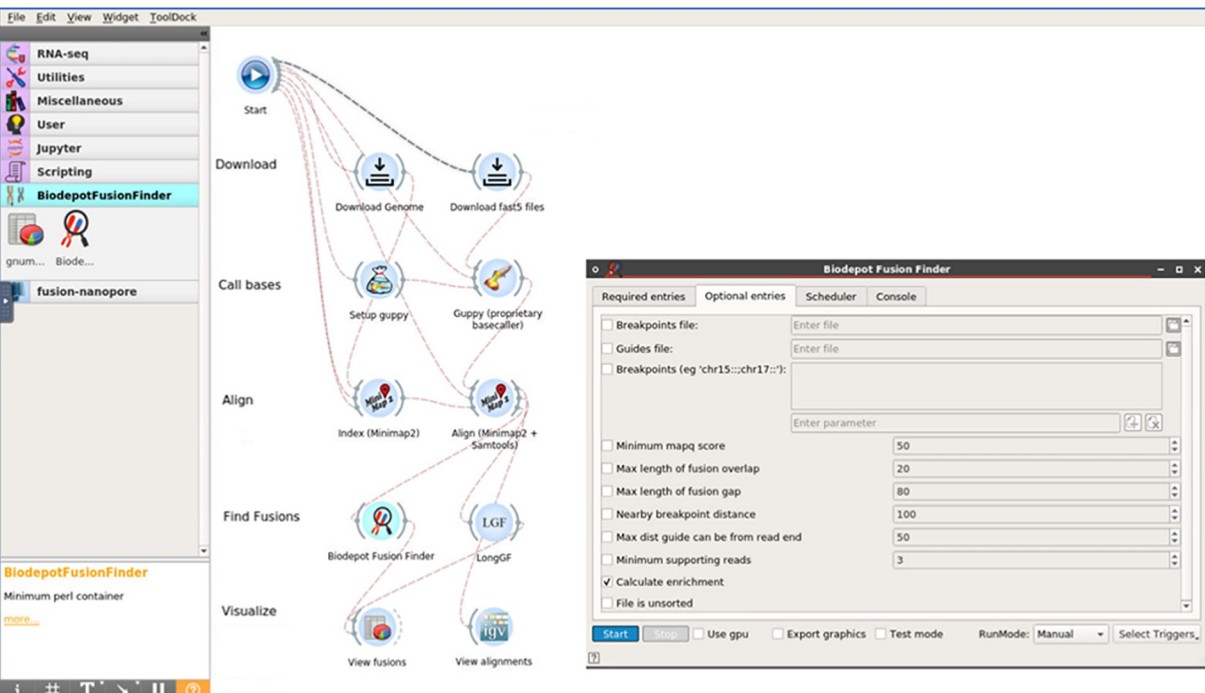

## B: Enrichment statistics computation output

| | A | B | C | D | E | F | G | H | I | J |
|---|---|---|---|---|---|---|---|---|---|---|
| 1 | Breakpoint | Gap/Overlap | Count | Nearby-count | ReadThru | NearbyReadThru | Breakpoint enrichment | Breakpoint nearby enrichment | Fraction on target | Fraction on target nearby |
| 2 | chr9:133607156;chr22:23632742 | 2 | 1 | 29 | 30 | 31 | 30.19597 | 875.68318 | 0.03333 | 0.93548 |
| 3 | chr9:133607152;chr22:23632739 | 4 | 1 | 29 | 29 | 31 | 30.19597 | 875.68318 | 0.03448 | 0.93548 |
| 4 | chr9:133607147;chr22:23632742 | 5 | 21 | 29 | 28 | 31 | 634.11541 | 875.68318 | 0.75000 | 0.93548 |
| 5 | chr9:133607164;chr22:23632742 | 5 | 1 | 29 | 31 | 31 | 30.19597 | 875.68318 | 0.03226 | 0.93548 |
| 6 | chr9:133607168;chr22:23632748 | 5 | 1 | 29 | 30 | 31 | 30.19597 | 875.68318 | 0.03333 | 0.93548 |
| 7 | chr9:133607158;chr22:23632736 | 8 | 1 | 29 | 31 | 31 | 30.19597 | 875.68318 | 0.03226 | 0.93548 |
| 8 | chr9:133607147;chr22:23632715 | 16 | 1 | 29 | 31 | 31 | 30.19597 | 875.68318 | 0.03226 | 0.93548 |
| 9 | chr9:133607177;chr22:23632748 | 17 | 1 | 29 | 30 | 31 | 30.19597 | 875.68318 | 0.03333 | 0.93548 |
| 10 | chr9:133607153;chr22:23632728 | 20 | 1 | 29 | 31 | 31 | 30.19597 | 875.68318 | 0.03226 | 0.93548 |
| 11 | Coverage | MaxBpFromEnd | GuideReads | Enrichment | | | | | | |
| 12 | 0.0331 | 50 | 212 | 6401.546 | | | | | | |

## C: IGV image confirming fusion

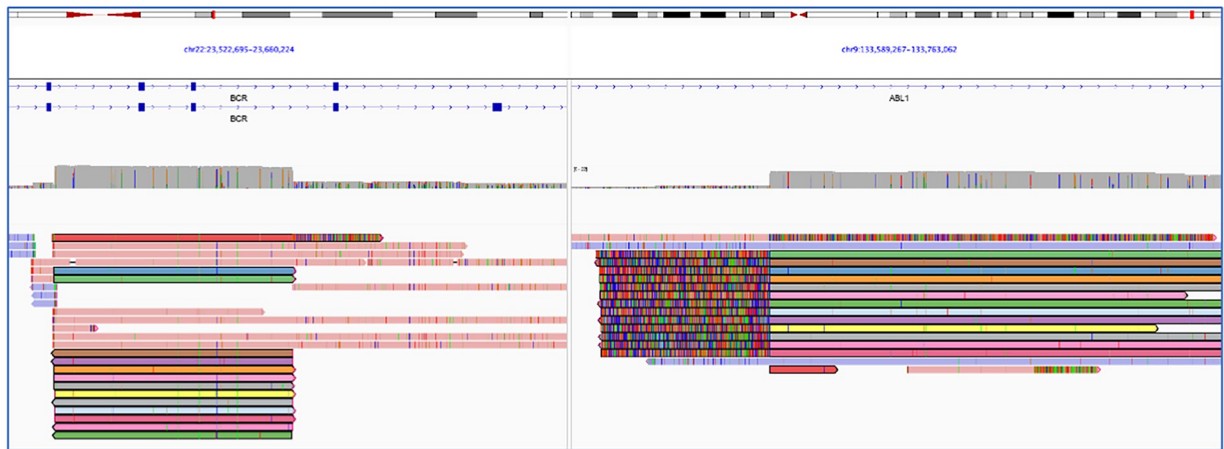

**Fig 2. Fusion detection bioinformatics workflow and output. Panel A:** Screenshot of Bwb workflow showing the display interface, including our custom Biodepot Fusion Finder (BFF) and LongGF widgets. **Panel B:** shows enrichment statistics as fusion enrichment and on-target enrichment. **Panel C:** alignment can be viewed on IGV based on a BAM file generated from minimap2. This case shows a t(9;22) *BCR::ABL1* fusion in a primary specimen. Reads spanning the breakpoint are colored with the same color in alignment to both genes.

### Benchmarking experiments

We performed empirical experiments to benchmark the sensitivity and runtime required to detect fusions reliably. Sequencing metrics, including quality scores and timestamps, are obtained for each sample from the sequencing summary text file, an output of the Guppy base caller. Fusions are detected using BFF and LongGF. BFF also provides specific fusion break-points. All samples are combined into a dictionary and separated by patient and cell line data. Plots are constructed for each category of data pertaining to the time to reach three fusions and the number of reads required to reach three fusions. Finally, the total number of fusion reads detected for each sample is compared between the BFF and LongGF, as shown in **Fig 2**.

### Enrichment assessment

Two enrichment metrics were computed by BFF and tracked for each sample. First, the *fusion-specific enrichment was* calculated with the formula [(number of fusion reads) / (mean coverage of the genome)]. Second, the *on-target enrichment* was calculated with the formula [(number of reads that originate from a guide RNA cut point that includes the region of the breakpoint) / (mean genome coverage)]. Reads originating from a guide RNA cut are distinguished by the guide sequence at the beginning of a read. As initial electrical signal data is generated by DNA passing through the nanopores (reads at the beginning of a strand) is error-prone, which affects base calling and, therefore the alignment of the reads so that the start sequence is often misaligned. Consequently, the BFF considers base pairs of the sequence near the start (default within 50 bp) of a read that aligns near the coordinates of a guide RNA cut site to have originated from a guide RNA cut. Specific reads cut by guides are manually confirmed. The allowed error intervals are customizable. Samtools v1.13 is used to sort and convert BAM files and determine the average coverage [31]. Picard CollectHsMetrics generated the unique base pairs mapped metric for each sample [32].

### Ethics statement

The Fred Hutchinson Cancer Center (FHCC) institutional review board approved this study (protocol numbers 1837, 1690, 547). All patients' samples used for this study were obtained through FHCC institutional review board protocols either as 1) discarded de-identified patient samples or 2) as consent patients collected into a biorepository. Patients enrolled through 1690 are consented, and patient consents are on file. Samples obtained through 547 and 1837 are considered non-human subjects and waived from the consenting process as they are deidentified materials that are discarded after clinical diagnosis has been rendered.

## Results

### Sample sequencing and enrichment

Details of the sample sequencing and enrichment metrics are included in **Tables 2 and 3**. A range of 0.04 Gb– 5.47 Gb (gigabases) of sequencing data were generated for each sample for an average mean coverage of the human genome of 0.32-fold (range: 0.01–1.66). We adopted standard quality metrics for nanopore workflows and tracked N50, a quality metric where half the reads are above this length (range 4.65kb-32.2kb) and median read length for total reads (range 0.87kb- 9.60kb). The percentage of reads aligned ranged from 72–96%. Fusion-specific enrichment was 135–837-fold for cell lines and 6–509-fold for patient samples. On-target enrichment was 849–5830-fold for cell lines and 535–3007-fold for patient samples.

**Table 3. Enrichment characteristics and comparison of fusion detection by BFF versus long GF.**

| Sample | Fusion Reads-Long GF | Fusion Reads—BFF | On-target Enrichment | Fusion Enrichment Spanning Breakpoint | Fusion Breakpoint coordinates | Disease Burden |
|---|---|---|---|---|---|---|
| K562 | 0 | 29 | 5830.00 | 797.50 | chr9:133607147; chr22:23632742 | Cell line |
| KU812 | 0 | 51 | 2108.33 | 467.50 | chr9:133643198; chr22:23632850 | Cell line |
| KCL22 | 5 | 171 | 2118.70 | 817.83 | chr9:133664205; chr22:23632116 | Cell line |
| NB4 | 3 | 3 | 848.57 | 141.43 | chr15:74326373; chr17:38502175 | Cell line |
| MV4;11 | 57 | 71 | 2698.93 | 836.79 | chr4:88009085; chr11:118353675 | Cell line |
| ME1 | 45 | 37 | 1305.49 | 134.18 | chr16:15815076; chr16:67132446 | Cell line |
| CML1 | 0 | 142 | 1563.91 | 509.35 | chr9:133709131; chr22:23632184 | CML-AP Relapse |
| CML2 | 0 | 4 | 2227.50 | 330.00 | chr9:133629166; chr22:23632505 | CML BC |
| CML3 | 0 | 0* | 3006.67 | NA | NA | CML-AP |
| CML4 | 0 | 14 | 557.10 | 49.68 | chr9:133636186; chr22:23634109 | CML-CP |
| CML5 | 0 | 8 | 1283.33 | 293.33 | chr9:133709142; chr22:23632178 | CML-AP Relapse |
| CML6 | 0 | 106 | 1539.12 | 142.56 | chr9:133582947; chr22:23633181 | CML BC |
| AML1 | 28 | 38 | 535.12 | 22.93 | chr9:20413972; chr11:118355951 | 90% blasts |
| AML2 | 0 | 0 | 649.62 | NA | NA | 20% blasts |
| APL1 | 0 | 0** | 1242.35 | NA | NA | 0% PB blasts, APL confirmed by FISH on BM sample |
| APL2 | 8 | 10 | 618.00 | 30.00 | chr15:74316531; chr17:38489068 | 80% blasts |
| APL3 | 8 | 10 | 872.90 | 106.45 | chr15:74316604; chr17:38493994 | 75% blasts |
| APL4 | 0 | 0*** | 1785.64 | NA | NA | 83% blasts |
| APL5 | 0 | 0*** | 2520.00 | NA | NA | 79% blasts |
| APL6 | 0 | 2**** | 650.66 | 6.23 | chr11:118354923; chr17:75328696 | 22% blasts on flow in a <5% marrow |

*Pt had very low BCR::ABL1 level that was only detected by qualitative PCR and not quantitative PCR (< 0.01%).

**Pt with confirmed BM disease, but the PB sample used for this assay had <1% blast count.

***Pt had an unusual fusion between a viral gene and *RARA* (reference: Sala-Torra, Blood Adv. Apr 14, 2022), however a human reference genome was used on BFF to generate this data and thus this fusion could not be detected.

****Patient was suspected to have APL, but subsequent cytogenetic results showed complex karyotype with an unexpected *KMT2A* fusion. This was also a challenged sample with an N50 < 5000bp.

CML = chronic myeloid leukemia; AML = acute myeloid leukemia; APL = acute promyelocytic leukemia, AP = accelerated phase, BC = blast crisis, PB = peripheral blood, BM = bone marrow; NA = not available; FISH = fluorescence *in-situ* hybridization.

## Concordance of fusion detection with clinical results

Concordance with expected results was 100% for cell lines. We detected the expected fusion in 6/6 cell lines (3 *BCR::ABL1*, 1 *PML::RARA*, 1 *CBFB::MYH11*, 1 *KMT2A::AFF1*). Breakpoint sequences detected for *BCR::ABL1* cell lines are the same as previously published [23]. We

correctly confirmed the presence or absence of fusions in 11/14 (78.5%) primary specimens, including both diagnostic and measurable residual disease (MRD) cases with a minimum of 3 reads However, one case (APL6) showed only two fusion reads and was not counted as confirmed. The three missed cases (CML3, APL1, and APL6 in Tables 1 and 2) were specimens with low disease burden, ~0%, 1%, and <5%. We detected *BCR*::*ABL1* in 5/6 cases, *PML*::*RARA* in 2/6 cases, and *KMT2A*::*MLLT3* and *CBFB*::*MYH11* in 1/1 case each. Orthogonal confirmation of the specific breakpoints was conducted for select patient cases via PCR (see **Fig 1**) and fusion breakpoints of cell lines were compared to published data (see **S1 Table**).

In three specimens from two patients with suspected APL, we could not detect *PML*::*RARA* but observed other findings. Clinical and laboratory details are listed in **Table 2**. For the first patient, two specimens, one bone marrow and one peripheral blood (APL4 and APL 5), yielded no *PML*::*RARA* fusion reads. This patient presented with an AML morphologically suggestive of APL and an isochromosome 17q without t(15;17) detected by karyotype. While fusion detection software did not detect a fusion, manual inspection showed an insertion in an intronic region of RARA with TTMV viral genome [21]. Patient APL6 (presented with 22% blasts on flow in a <5% marrow which on unblinding showed a complex karyotype with t (11;17)(q23;q25), including the *KMT2A* gene. Two reads with *KMT2A*::*SEPTIN9* fusion were detected in a suboptimal but acceptable run (N50 < 5000bp; on-target enrichment 650.66 fold), confirming the lack of t(15;17) or *PML*::*RARA* fusion, but the threshold was below the requisite three reads to confirm the *KMT2A*::*SEPTIN9* fusion.

## Comparison of fusion detection tools

A comparison of two bioinformatic workflows using different fusion detection methods, namely LongGF vs. Biodepot Fusion Finder (BFF), was conducted. These two workflows are depicted in **Fig 2**. Additionally, BFF computes the fusion enrichment and on-target enrichment statistics that are summarized in **Table 2**. In most cell line and primary specimens, LongGF has problems detecting BCR::ABL1 and misses fusion reads that BFF identifies. Using the BFF, the average total sequencing and data processing time to 3 reads with fusions in the cell line experiments was 42.75 minutes (range: 18.87–77.65 min) and 188 minutes in the primary specimens where three reads were detected (range: 32–654 min) [see **Fig 3**, and **S1 Table**]. Cell line experiments took an average of 11,711 reads to identify three fusion reads (range: 1,321–43,326 reads) and 10,273 reads in primary specimens (range: 1,790–24,999 reads) for confirmation of the fusion calls. BFF provides precise breakpoints in each read in addition to the number of breakpoints found nearby (see **Table 3**). In contrast, LongGF only detects and provides breakpoint coordinates only when there are enough supporting reads with the same breakpoint.

## Comparison of Flongles and MinION flow cells

In five of our experiments (2 cell lines, three primary specimens), we used Flongles; in the other experiments, we used MinION flow cells. The performance of the affordable Flongles was inferior to the MinION flow cells, with lower expected average data output from the Flongles (based on manufacture expectations of ~3GB), the N50 and median read length were smaller in Flongle reads (21,614 vs. 19,166 and 6058 vs. 5250 respectively), and significantly, the median Phred score for Flongle reads was lower than that for the MinION flow cells (9.2 vs. 11.88). Despite the worse performance of the Flongles, fusions were detected in 2 of 2 cell lines, and 2 of the three experiments with primary specimens with at least 8 and 14 reads confirming the fusion, and the specimen without fusion confirmation was CML3 with pancytopenia and low disease burden.

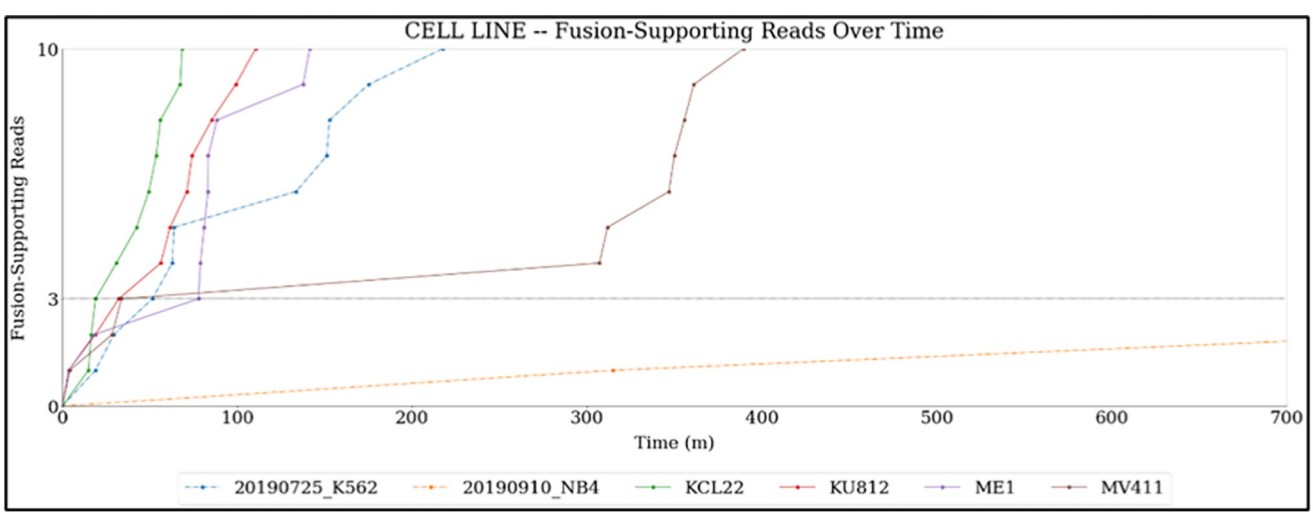

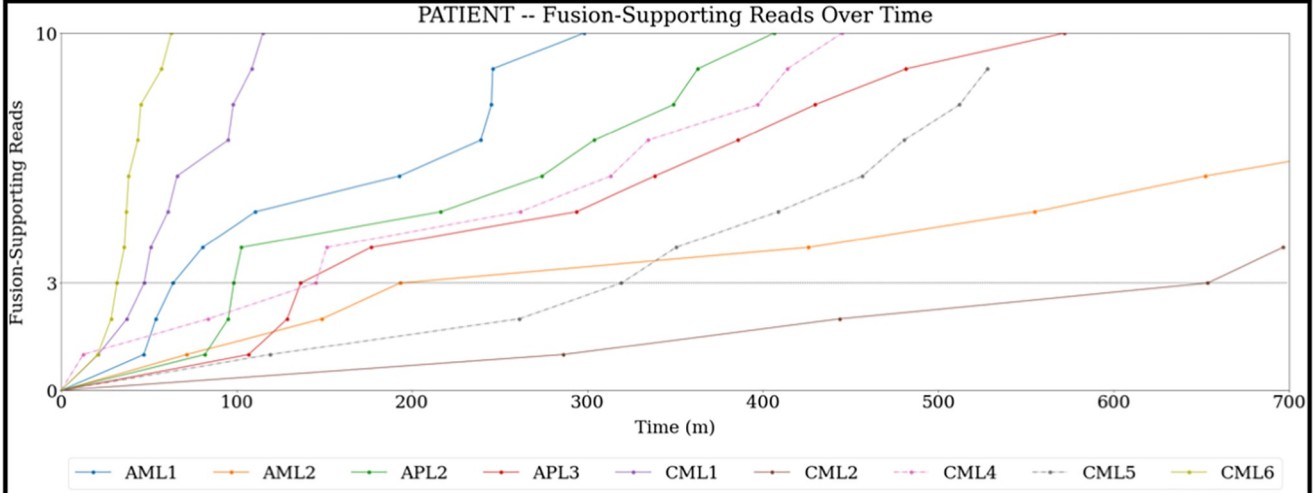

**Fig 3. Time required to obtain 3 fusion supporting reads.** The top panel shows the time to 3 fusion supporting reads seen in the cell lines samples. The bottom panel shows time to 3 fusion supporting reads seen in patient samples. Details of specific times are listed in **S1 Table**.

## Discussion

Our laboratory has been providing supportive molecular cancer diagnoses to chronic myeloid leukemia patients through the Spot on CML program with the MAX foundation [33] and recognizes the need to provide testing that is more portable with a faster turnaround time, tests that can be done closer to the patients, and platforms that can assay several targeted genes simultaneously. In response, we developed a rapid assay to detect fusion genes in blood or marrow samples in less than 8 hours. The fastest time to achieve three fusion reads was 5 hours on a portable sequencing device with an accessible cloud-based data analysis workflow. This was accomplished by combining CRISPR-Cas9 enrichment during library preparation, nanopore long-read sequencing, and a cloud-based data analysis pipeline with a novel BFF program developed and optimized for finding fusion reads and breakpoints. RNA guides were designed to target genes involved in recurrent fusions in myeloid malignancies and used to enrich an amplification-free library preparation over 1600-fold. Our modular and containerized pipeline in Bwb allows users to efficiently process raw FAST5 data on the cloud through

an accessible graphical user interface allowing for a very fast analysis step (average ~4.5 minutes for base calling, alignment, and fusion detection). To improve fusion calling, we developed the custom BFF that allows users to identify fusion reads not detected by LongGF. We successfully confirmed published genomic breakpoints in our series of cell lines and archival patient samples, including both diagnostic and follow-up samples, to test feasibility in confirming both common and novel fusions over a range of tumor burdens. Our study includes 14 patient specimens and demonstrates the usability of this method in primary specimens with 2 micrograms of DNA.

An advantage of the CRISRP-Cas9 based enrichment protocol is that it allows targeting multiple common leukemia fusion genes by pooling multiple guide RNAs. Fusions are particularly well suited for this method as they have large gene segments that aid in alignment despite sequencing errors and wide variation where translocation breakpoints may occur. Other labs have employed similar but different methods that detected fusions by targeting one partner gene in the fusion [24]. In contrast, our assay targets both fusion partners; thus our approach allows for an expanded capability to detect known and novel fusions, such as in the case AML1 with t(9;11), when guides are designed to target t(4;11). However, in 2/14 patients, the assay did not detect fusion genes because both cases had a very low amount of fusion target. In one case (CML3) because the BCR::ABL1 was <0.01%, the other was in the context of a very hypocellular sample.

Our work differs from standard RNA-based fusion detection assays and instead interrogates single molecules of DNA rapidly and accurately to detect specific translocation breakpoints. Long-read sequencing technologies, like Nanopore (Oxford Nanopore Technologies, Oxford, U.K.), allow the sequencing of unamplified, long unbroken fragments of DNA which are more likely to span a breakpoint. This genomic breakpoint identification has potential clinical utility for personalized disease monitoring when CML patients are on TKI therapy and suppressing RNA transcription [34], targeting DNA as a monitoring target may be more robust and reproducible since DNA is stable and present in constant numbers [35]. This targeting DNA rather than RNA has higher tolerance for challenged samples, and testing tissues with poor preservation is possible, which would be especially beneficial for LMICs and remote areas that may not have immediate access to a pathology laboratory or even refrigeration. However, genomic breakpoints in *BCR::ABL1* are unique to individual patients requiring patient-specific breakpoint characterization [35–39] as *ABL1* breakpoints occur over an expansive region of about 150Kb, making this an arduous endeavor previously involving multiple primer sets and Sanger sequencing [40]. Our method allows a single approach spanning the *BCR* and *ABL1* breakpoint regions without the use of multiple primers and PCR reactions. Once the breakpoint is known, the sensitivity of DNA-based qPCR can be as low as $10^{-7}$ [41, 42]. Specimens CML1 and CML5 are a BM and PB (respectively) obtained from the same patient and demonstrate high fidelity in confirming genomic breakpoints and the ability to use patient-specific primers for personalized MRD monitoring.

The advantages of long-read sequencing over current clinical diagnostic assays are speed and the relatively low complexity of the assay when compared to cytogenetics and targeted NGS panels. While long-read sequencing results could potentially have a turnaround time (TAT) of less than 24 hours, full karyotype analysis TAT is generally longer, with the fastest times at days to a week, and most targeted NGS panels require ~7–10 days from the start of processing to the result report. The nanopore sequencing generates data that can be simultaneously analyzed by our GPU-enabled data analytic pipeline in the Bwb interface, which resides on the cloud to help interpret and reliably identify fusion reads within 5000 seconds (<2 hours) of computational time in most specimens. Building on our experience [20, 24], we used three reads as a threshold for fusion confirmations. With the simultaneous sequencing

and data analysis workflow described here, three sequences are detected in an average of 3 hours and 7 minutes (fastest at 30 minutes) in the nine patient specimens where a fusion could be confirmed. Consequently, a diagnostic result with a precise fusion breakpoint with three fusion supporting reads would be possible on the same day.

Cost was a consideration in developing this assay, as the total cost per assay must be sustainable for patients in LMICs. Five specimens were sequenced on the more affordable Flongle device, which has lower sequencing capabilities with fewer pores and channels but costs around $90 per sequencer. Although the library preparation reagent costs for these early proof of concept experiments were the same (approximately $100), further optimization could lower costs for the Flongle devices. On the Flongles, three fusion reads were reached in 4/5 specimens (2 cell lines, three patient samples). The sequencing quality is significantly lower on the Flongles (median Phred 9.2 in Flongles vs. 11.9 in MinION flow cells). However, fusions were detected in all samples with tumor burden above 5%. We predict that additional optimizations of the CRISPR library guides will increase efficiency and on-target fusion reads, which can overcome reduced numbers of pores on the Flongles. However, in our experience, the pores on the Flongles have poorer viability and generate more read errors demonstrable by lower Flongle Phred quality on average that result in more challenging alignment. Therefore, a study of errors specific in data sequenced by the Flongle compared to the MinION flow cells is needed to understand differences in data generated. New generations of flow cells and chemistry kits from Oxford Nanopore and alternative synthetic nanopore devices could overcome these challenges. To achieve a cheaper version of our fast and portable nanopore fusion assay, more optimization on the Flongle devices will need to be developed, and our preliminary work demonstrates that an even lower cost assay is possible.

## Conclusion

We demonstrate the feasibility of using single molecule long-range sequencing assay to detect fusion genes in patients with heme malignancies (AML, CML and APL). Inherent characteristics of fusions make this assay a promising, cost-effective tool for rapid detection of recurrent fusions that 1) require previous knowledge of only one of two target genes as guides can capture an unknown partner gene, 2) has a rapid TAT (8 hours in 80% of samples) when multiplexing different assays and used with the specific data analysis and fusion detection tools, 3) can precisely map translocation genomic breakpoints that allow for development of personalized markers for disease monitoring, and 4) can potentially allow discovery of novel/different fusion partners. Our proof-of-concept work described in this study shows that a low-cost, portable fusion cancer diagnostic device with an integrated cloud-based on-demand analysis workflow can be implemented in LMICs. Furthermore, no expensive lab investment or computing infrastructures is needed, thus bridging the gap between diagnosis and tailored treatment administration.

## Supporting information

**S1 Table. Times and total reads before 3 fusion reads could be confirmed in cell lines and patient samples.**
(DOCX)

**S1 Fig. Genomic BCR-ABL1 breakpoint identified in a primary specimen by long-read sequencing.** A) Schematic representation of the breakpoint captured with our amplification-free enrichment protocol and long-read sequencing. Reads capturing the breakpoint are up to approximately 23Kb long and represented in the graph by the red line: 2KB in BCR,

chromosome 22, and 21KB in ABL1, chromosome9. B) Nanopore sequence IGV visualization showing the alignment of multiple sequences with ABL1 (red arrow). The fragment marked by the blue arrow does not align with ABL1 and corresponds to BCR. C) Agarose gel electrophoresis (2% agarose) of PCR amplified product using patient specific primers designed after breakpoint detection. The gel shows the patient sample in lane 1 and a BCR-AB1 cell line (KCL22) in lane 2. D) Confirmation on the patient specific breakpoint by Sanger sequencing. (TIFF)

## Acknowledgments

The authors want to thank Dr. Phillip E. Starshak, Kaiser Permanente, Oakland, CA for procuring some of the specimens.

## Author Contributions

**Conceptualization:** Olga Sala-Torra, Cecilia C. S. Yeung.

**Formal analysis:** Olga Sala-Torra, Shishir Reddy, Ling-Hong Hung, Lan Beppu.

**Funding acquisition:** Jerald Radich, Ka Yee Yeung, Cecilia C. S. Yeung.

**Investigation:** Olga Sala-Torra, Ling-Hong Hung, Lan Beppu, Cecilia C. S. Yeung.

**Methodology:** Olga Sala-Torra, Ling-Hong Hung, Cecilia C. S. Yeung.

**Project administration:** Cecilia C. S. Yeung.

**Resources:** David Wu, Jerald Radich, Cecilia C. S. Yeung.

**Software:** Shishir Reddy, Ling-Hong Hung.

**Supervision:** Jerald Radich, Ka Yee Yeung, Cecilia C. S. Yeung.

**Validation:** Olga Sala-Torra, Ka Yee Yeung, Cecilia C. S. Yeung.

**Visualization:** Olga Sala-Torra, Shishir Reddy, Ling-Hong Hung.

**Writing – original draft:** Olga Sala-Torra, Shishir Reddy, Ka Yee Yeung, Cecilia C. S. Yeung.

**Writing – review & editing:** Olga Sala-Torra, Shishir Reddy, Ling-Hong Hung, David Wu, Jerald Radich, Ka Yee Yeung, Cecilia C. S. Yeung.

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
