## [Decision Letter · Decision Letter 0]

13 Jan 2023

PGPH-D-22-01725

Rapid detection of myeloid neoplasm fusions using single-molecule long-read sequencing

Dear Dr. Yeung,

Thank you for submitting your manuscript to PLOS Global Public Health. After careful consideration, we feel that it has merit but does not fully meet PLOS Global Public Health’s publication criteria as it currently stands. Therefore, we invite you to submit a revised version of the manuscript that addresses the points raised during the review process.

The manuscript has been evaluated by two reviewers, and their comments are available below.

The reviewers have raised a number of major concerns regarding the reporting of the study design, methods, and results. In particular, Reviewer #2 has concerns about whether all of the stated conclusions are supported by the data reported in the study. 

Please carefully revise your manuscript to address each of the comments raised.

We look forward to receiving your revised manuscript.

Kind regards,

Hugh Cowley

Staff Editor

Journal Requirements:

1. Our staff editors have determined that your manuscript is likely within the scope of our Diagnostics in Global Health Call for Papers. This editorial initiative is headed by a team of Guest Editors for PLOS GPH: Senjuti Saha (Child Health Research Foundation, Bangladesh) and Titus Divala (Public Health Scotland, University of Glasgow and University of Malawi College of Medicine). The Collection will encompass a diverse range of research articles about diagnostics in global health, including innovation and deployment of point of care diagnostics; subsets of diagnostics related to infectious diseases, chronic diseases and injuries; policies related to and regulation of diagnostics; supply chain issues; and the affordability, accessibility, and availability of essential diagnostics.  Additional information can be found on our announcement page: https://collections.plos.org/call-for-papers/diagnostics-in-global-health/

If you would like your manuscript to be considered for this collection, please let us know in your cover letter and we will ensure that your paper is treated as if you were responding to this call.  Please note that being considered for the Collection does not require additional peer review beyond the journal’s standard process and will not delay the publication of your manuscript if it is accepted by PLOS GPH. If you would prefer to remove your manuscript from collection consideration, please specify this in the cover letter.

2. Please send a completed 'Competing Interests' statement, including any COIs declared by your co-authors. If you have no competing interests to declare, please state "The authors have declared that no competing interests exist". Otherwise please declare all competing interests beginning with the statement "I have read the journal's policy and the authors of this manuscript have the following competing interests:"

3. Please amend your detailed Financial Disclosure statement. This is published with the article. It must therefore be completed in full sentences and contain the exact wording you wish to be published.

a. State what role the funders took in the study. If the funders had no role in your study, please state: “The funders had no role in study design, data collection and analysis, decision to publish, or preparation of the manuscript.”

b. If any authors received a salary from any of your funders, please state which authors and which funders.

3. We ask that a manuscript source file is provided at Revision. Please upload your manuscript file as a .doc, .docx, .rtf or .tex.

4. Please provide separate figure files in .tif or .eps format.

5. We do not publish any copyright or trademark symbols that usually accompany proprietary names, eg (R), (C), or TM  (e.g. next to drug or reagent names). Please remove all instances of trademark/copyright symbols throughout the text, including R on page 6.

6. We notice that your supplementary figures are uploaded with the file type 'Figure'. Please amend the file type to 'Supporting Information'. Please ensure that each Supporting Information file has a legend listed in the manuscript after the references list.

7. In the online submission form, you indicated that "All de-identified data without PHI in this study is available and can be provided upon request to the corresponding author(s).". All PLOS journals now require all data underlying the findings described in their manuscript to be freely available to other researchers, either 1. In a public repository, 2. Within the manuscript itself, or 3. Uploaded as supplementary information.

Additional Editor Comments (if provided):

Reviewers' comments:

Reviewer's Responses to Questions

**Comments to the Author**

1. Does this manuscript meet PLOS Global Public Health’s publication criteria? Is the manuscript technically sound, and do the data support the conclusions? The manuscript must describe methodologically and ethically rigorous research with conclusions that are appropriately drawn based on the data presented.

Reviewer #1: Yes

Reviewer #2: No

2. Has the statistical analysis been performed appropriately and rigorously?

Reviewer #1: N/A

Reviewer #2: N/A

3. Have the authors made all data underlying the findings in their manuscript fully available (please refer to the Data Availability Statement at the start of the manuscript PDF file)?

Reviewer #1: Yes

Reviewer #2: No

4. Is the manuscript presented in an intelligible fashion and written in standard English?

Reviewer #1: Yes

Reviewer #2: Yes

5. Review Comments to the Author

Reviewer #1: Excellent and well-designed study.

I would like to kindly inquire about the following:

- Why wasn't RNUX1-RUNX1T1 tested?

- What is the claimed minimum %read alignment adopted for "excellent" quality?

- Were the discrepant cases ever retested using RNA instead of DNA as a starting genomic material?

- Kindly check grammar mistakes throughout the manuscript

Reviewer #2: Sala-Torra et al applied PCR-free CRISPR-Cas9 targeted DNA nanopore sequencing to detect gene fusions from 6 cell lines and 14 clinical samples. This reviewer has the following reservations regarding study design, methods, results and conclusions.

1. Study design

- (lines 150-154) Guide RNA are designed in both forward and reverse directions of each entire gene? Does "MYH11-CBFB" mean that the canonical driver fusion CBFB-MYH11 is also covered? Are PML-RARA bcr1/bcr2/bcr3 variants all covered? Is the BCR-ABL1 p210 covered but not p190?

- (Table 1) Sample "CML3" has p190 isoform but the assay is designed to detect p210 (line 153). All 3 BCR-ABL1 cell lines have p210 isoform as well (K562:b3a2, KU812:b3a2, KCL22:b2a2). False negative detection can be attributed to assay design instead of "low disease burden" (line 298).

- (lines 335-339) Flongle and MinION output comparison (data output yield, read length, phred score) are not based on parallel run using the same input DNA and sequencing library. Input DNA quality and sequencing library preparation batch effect could be confounding factors.

- If CML is suspected, will RT-PCR or GeneXpert cartridge be cheaper and quicker alternative to detect BCR-ABL1 fusion for low/middle income countries?

2. Methods

- (line 168) Is a new flow cell is used for each sample? Same run time of 72 hours? Which Guppy basecaller version and basecalling model?

- Is the BFF source available for readers to implement in their own lab?

3. Results

- (Table 2) Results table and paragraph are difficult to read and interpret. For each sample (cell line, clinical sample), what is the expected result, true positive / false negative fusion detected, absolute number of reads, and how many false positive fusion calls >= 3 reads. Enrichment ratio is a good sample-level QC metric. However, for a given fusion, e.g. BCR-ABL1, absolute number of reads (fusion spanning reads for true positive, average read count for false negative) is needed for interpretation. In fact Table 2 is incompletely shown in the PDF file and the right hand side portion is missing.

- (line 323) Sample bioinformatic analysis time 188 minutes is inconsistent with discussion (line 358: 4.5 minutes)

- Fusion genes should be consistently presented using conventional names (MLL-AF4) or latest nomenclature (KMT2A-AFF1) (line 293) but not a mix (KMT2A-AF4) (line 42). More examples include KMT2A-SEPTIN9 or MLL-SEPT9 but not KMT2A-SEPT9 (Line 313)

- (Table 2) Sample "CML6" what is "in" BFF fusion read count?

- (Table 2 and line 168) Flongle is also flow cell. Therefore, "flowcell" is ambiguous and confusing. Flongle versus MinION flowcell may help to differentiate.

4. Conclusion

- (line 434) Does not require previous knowledge of target? but the presented assay is targeted sequencing not whole genome sequencing

- (line 434) TAT: Is sample extraction and library preparation and bioinformatics time is included?

- (line 436) Since data is not shown for genomic breakpoint sequence nor are the primers available for review, it remains to be shown how one can precisely map translocation genomic breakpoint by this assay.

6. PLOS authors have the option to publish the peer review history of their article (what does this mean?). If published, this will include your full peer review and any attached files.

**Do you want your identity to be public for this peer review?** For information about this choice, including consent withdrawal, please see our Privacy Policy.

Reviewer #1: **Yes: **RAMI MAHFOUZ MD,MPH,IFCAP

Reviewer #2: No

---

## [Decision Letter · Decision Letter 1]

10 May 2023

PGPH-D-22-01725R1

Rapid detection of myeloid neoplasm fusions using single-molecule long-read sequencing

Dear Dr. Yeung,

Thank you for submitting your manuscript to PLOS Global Public Health. After careful consideration, we feel that it has merit but does not fully meet PLOS Global Public Health’s publication criteria as it currently stands. Therefore, we invite you to submit a revised version of the manuscript that addresses the points raised during the review process.

The manuscript has been evaluated by three reviewers, and their comments are available below.

The reviewers have made several requests for clarifications.

Could you please revise the manuscript to carefully address the concerns raised?

We look forward to receiving your revised manuscript.

Kind regards,

Steve Zimmerman, PhD

PLOS Staff Editor

Journal Requirements:

Additional Editor Comments (if provided):

Reviewers' comments:

Reviewer's Responses to Questions

**Comments to the Author**

1. If the authors have adequately addressed your comments raised in a previous round of review and you feel that this manuscript is now acceptable for publication, you may indicate that here to bypass the “Comments to the Author” section, enter your conflict of interest statement in the “Confidential to Editor” section, and submit your "Accept" recommendation.

Reviewer #2: (No Response)

Reviewer #3: All comments have been addressed

Reviewer #4: (No Response)

2. Does this manuscript meet PLOS Global Public Health’s publication criteria? Is the manuscript technically sound, and do the data support the conclusions? The manuscript must describe methodologically and ethically rigorous research with conclusions that are appropriately drawn based on the data presented.

Reviewer #2: Yes

Reviewer #3: Yes

Reviewer #4: Yes

3. Has the statistical analysis been performed appropriately and rigorously?

Reviewer #2: N/A

Reviewer #3: Yes

Reviewer #4: Yes

4. Have the authors made all data underlying the findings in their manuscript fully available (please refer to the Data Availability Statement at the start of the manuscript PDF file)?

Reviewer #2: Yes

Reviewer #3: Yes

Reviewer #4: Yes

5. Is the manuscript presented in an intelligible fashion and written in standard English?

Reviewer #2: Yes

Reviewer #3: Yes

Reviewer #4: Yes

6. Review Comments to the Author

Reviewer #2: The effort of the authors to meticulously revise the manuscript is appreciated.

I wish to further clarify the following 2 points:

1. Patient CML3: this patient is diagnosed as long standing p190 CML but now the disease is MDS/MPN and the BCR::ABL1 fusion transcript is at a low level. Is the patient on TKI treatment and there is clonal escape to another genetic driver, or is this a therapy-related secondary disease?

2. Patient APL6: please clarify the meaning of 22% blasts in a < 5% marrow.

Reviewer #3: It is a well-written paper to describe a long-read sequencing DNA assay designed with CRISPR guides to select and enrich for recurrent leukemia fusion genes, that does not need a priori knowledge of the abnormality present. The authors have developed an amplification-free CRISPR-Cas9 targeted enrichment sequencing protocol using Nanopore MinION flow cell and Flongles to detect fusions relevant to the diagnosis and classification of CML and AMLs.

One of the questions that I believe arises after reading the paper is: is there a specific tumor fraction above which this technology works? can a dilution series experiment be performed using a high tumor fraction sample or cell line to understand how the technology performs as a function of tumor fraction?

Minor comments:

The font sizes maybe different before and after line 54 (lines 56-64)

higher quality images would make the figures easier to read. it’ll be great if this can be updated

Reviewer #4: I must admit this describes a well thought experiment. The manuscript is well detailed. I however have a couple of queries from the different section of the manuscript.

Methods

Line 51: The manuscript mentions that theprotocol described takes could be performed in only 8hours. I however realize that the sequencing run on the nanopore device oftentimes run overnight (from your response to another reviewer) and also from my experience with nanopore know most run requires a minimum of about 12 hours to attain good read depth.

Line 165: The word “Does not require previous knowledge of target” is misleading since the CRISPR guide system used is specific to concentrate known (specific) fusion regions.

Line 197: Sanger sequencing was used to confirm the BCR-ABL breakpoint, what method was used to verify the other identified breakpoints?

The possibility of multiplexing sequencing run with nanopore is another advantage of the described assay which I think should be highlighted.

Was this experiment repeated to ascertain the precision of the described assay/protocol?

7. PLOS authors have the option to publish the peer review history of their article (what does this mean?). If published, this will include your full peer review and any attached files.

**Do you want your identity to be public for this peer review?** For information about this choice, including consent withdrawal, please see our Privacy Policy.

Reviewer #2: No

Reviewer #3: No

Reviewer #4: **Yes: **Noah O. Bukoye

---

## [Decision Letter · Decision Letter 2]

18 Jul 2023

Rapid detection of myeloid neoplasm fusions using single-molecule long-read sequencing

PGPH-D-22-01725R2

Dear Dr. Yeung,

We are pleased to inform you that your manuscript 'Rapid detection of myeloid neoplasm fusions using single-molecule long-read sequencing' has been provisionally accepted for publication in PLOS Global Public Health.

Best regards,

Julia Robinson

Staff Editor

Reviewer Comments (if any, and for reference):

Reviewer's Responses to Questions

**Comments to the Author**

1. If the authors have adequately addressed your comments raised in a previous round of review and you feel that this manuscript is now acceptable for publication, you may indicate that here to bypass the “Comments to the Author” section, enter your conflict of interest statement in the “Confidential to Editor” section, and submit your "Accept" recommendation.

Reviewer #2: All comments have been addressed

Reviewer #3: All comments have been addressed

2. Does this manuscript meet PLOS Global Public Health’s publication criteria? Is the manuscript technically sound, and do the data support the conclusions? The manuscript must describe methodologically and ethically rigorous research with conclusions that are appropriately drawn based on the data presented.

Reviewer #2: Yes

Reviewer #3: Yes

3. Has the statistical analysis been performed appropriately and rigorously?

Reviewer #2: N/A

Reviewer #3: Yes

4. Have the authors made all data underlying the findings in their manuscript fully available (please refer to the Data Availability Statement at the start of the manuscript PDF file)?

Reviewer #2: Yes

Reviewer #3: No

5. Is the manuscript presented in an intelligible fashion and written in standard English?

Reviewer #2: Yes

Reviewer #3: Yes

6. Review Comments to the Author

Reviewer #2: I have no further comment.

Reviewer #3: This is a well written paper and all the reviewer comments have been answered to satisfaction. The data isn't available yet but the authors mention that it will be available after the manuscript is accepted.

7. PLOS authors have the option to publish the peer review history of their article (what does this mean?). If published, this will include your full peer review and any attached files.

**Do you want your identity to be public for this peer review?** For information about this choice, including consent withdrawal, please see our Privacy Policy.

Reviewer #2: No

Reviewer #3: No
